# Consequences of Sisyphean Efforts: Meaningless Effort Decreases Motivation to Engage in Subsequent Conservation Behaviors through Disappointment

**Katarzyna Byrka** [1,*,†] ⬤, **Katarzyna Cantarero** [1] ⬤, **Dariusz Dolinski** [1] ⬤ and **Wijnand Van Tilburg** [2] ⬤

[1] Social Behavior Research Center, Faculty of Psychology, SWPS University of Social Sciences and Humanities, 53-238 Wroclaw, Poland; kcantarero@swps.edu.pl (K.C.); ddolinski@swps.edu.pl (D.D.)

[2] Department of Psychology, University of Essex, Wivenhoe Park Colchester CO4 3SQ, UK; wijnand.vantilburg@essex.ac.uk

\* Correspondence: kbyrka@swps.edu.pl

† Current address: SWPS University of Social Sciences and Humanities, Ostrowskiego 30b, 53-238 Wroclaw, Poland.

**Abstract:** This paper explores the consequences of engaging in conservation efforts that later appear purposeless. Specifically, we tested the model in which disappointment lays at the root of decreased motivation in such situations. In Studies 1 and 2, participants (*n* = 239 and *n* = 283) imagined that they had recycled plastic bottles for a week and that an assistant had collected their garbage in either separate bags (meaningful condition) or only one bag (meaningless condition). Half of participants imagined that they had put plastic bags and screw caps into separate containers (low-effort condition), the other half imagined that they had torn off the label bands (high-effort condition). In Study 3, a longitudinal field experiment, participants (*n* = 286) took part in a real situation that followed the procedure from Studies 1 and 2. Altogether, we confirmed the moderating effect of effort on relationship between meaninglessness and motivation through experienced disappointment. We discuss consequences of efforts wasted for beliefs, intentions and behaviors affording sustainable solutions.

**Keywords:** meaningfulness; effort; disappointment; conservation behaviors; behavioral intention

## 1. Introduction

This research was inspired by real-life observations and media coverage. Local press has been reporting situations of sanitation workers commingling already recycled garbage with refuse and putting everything together into one truck often due to structural shortcomings (see, e.g., Philadelphia Inquirer [1], Toronto CBS [2] or Gazeta Wyborcza Lodz [3]). Individuals who witnessed garbage put altogether despite being previously sorted out question the sense of recycling and report to be discouraged [2]. We expect that conservation endeavors that appear meaningless have affective and motivational consequences that are detrimental for people's future engagement and participation in sustainable activities. The objective of this research is to scrutinize systematically situations in which conservation efforts become Sisyphean efforts.

According to Greek mythology, Sisyphus—the dishonest king of Ephyra—recurrently engaged in meaningless behavior. Each dawn, Sisyphus rolled a huge boulder up a hill, only for the boulder to roll back down just before he reached the top. The king had no choice but to repeat this behavior as he was cursed by the gods. People also encounter situations such as this on a daily basis, albeit not to the same extreme. Examples of such Sisyphean efforts—that is, efforts that appear meaningless and do not lead to a purposeful goal—include cleaning up after toddlers, pulling weeds or, as sometimes reported by the press, recycling waste into separate containers that is later collected by one truck.

Despite the existence of age-old examples of Sisyphean efforts, we lack a deep understanding of the affective and motivational consequences of engaging in effortful meaning-

less tasks. Anecdotal evidence from press reports suggests that Sisyphean efforts may result in a decrease in intrinsic motivation. To date, research has shown that the meaningfulness or meaninglessness of a task affects the quality and quantity of a person's performance (e.g., [4,5]). It is also known that extreme situations in which recurrent efforts do not lead to any expected outcomes may result in learned helplessness, characterized by decreased motivation to act (e.g., [6]).

To the best of our knowledge, the synergistic effect of the meaninglessness of a task and the effort put into the pursuit has not yet been explored, nor have its affective roots been given enough attention. The literature does not specify the types of emotions typically evoked in such situations. Based on previous research in the field of decision-making, we expected that disappointment characterized by powerlessness and perceived lack of agency may play a role of a significant mediator. This emotion arises in situations in which a desired outcome or goal is not achieved by an individual due to external circumstances [7].

Our specific objective was to determine whether putting effort into a meaningless conservation tasks decreases people's motivation to act pro-environmentally at the next occasion. Specifically, we explored various indices of motivation to engage in conservation efforts, such as perceived sense in continuing recycling, intention to continue recycling, intention to engage in related pro-environmental behaviors and actual behavior. We proposed that disappointment drives the effect of meaningless effort on motivation to engage in subsequent behaviors. In particular, we hypothesized that the demotivating effect of meaningless effort is mediated by experienced disappointment.

## 1.1. Motivational Consequences of Meaninglessness

According to Baumeister's conceptualization [8], meaning is defined as the "mental representations of possible relationships among things, events and relationships" (p. 15). Meaning is the term used to denote what connects objects, giving sense to the relationship that may unite them. A given situation may be meaningful in an epistemic sense or in a teleological sense [9].

Understanding how things and objects relate to one another is foundational for effective social functioning [10]. The process of linking actions to their consequences allows individuals to predict what is expected by others and how to pursue important goals. In line with the meaning maintenance model (MMM), people organize their experiences into a meaningful framework over the courses of their lifespans [10]. Meaningless situations violate this framework of expected relationships between things, events and objects [11].

Perceived meaninglessness affects motivation to engage in behaviors (e.g., [12,13]). Ariely et al. [4] showed that, when a task is meaningless, individuals are less prone to perform it. In one study, participants were asked to mark the letters "SS" on sheets with strings of letters. Later, their work was destroyed (shredded), ignored (put into a large file of sheets) or acknowledged (signed by a participant). In the shredded and ignored conditions, the number of completed sheets with marked letters was significantly lower than in the acknowledged condition. In another study, participants constructed Lego robots in exchange for monetary gain. In the meaningful condition, the robots were placed, one after another, in front of the participants after completion. In the meaningless condition, the robots were deconstructed in front of the participants. Ariely et al. [4] found that people constructed fewer robots in the meaningless condition.

In a similar study, Chandler and Kapelner [5] asked mTurk employees to mark infected cells on pictures in exchange for financial remuneration. Depending on the manipulation, participants either received the information that their work had served a purpose (meaningful condition), received no additional information (control condition) or were informed that their work was only a test trial and would not be recorded (meaningless condition). Participants perceived the meaningful condition as more meaningful than the control condition, but their perception of the meaningless condition did not differ from that of the control condition. Participants marked more cells in the meaningful condition

than in the control condition, but the number of marked cells did not differ significantly between the meaningless and control conditions.

### 1.2. Motivational Consequences of Meaningless Effort

Meaningless effort is understood as expenditure of energy, time or other resources that does not conclude in an expected outcome and is thus perceived as having no sense or not serving a purpose. To our knowledge, no previous studies have directly tested how meaninglessness and effort interact in affecting motivation. However, the existing literature, which primarily focuses on effort, allowed us to have some predictions. In principle, people often invest effort into causes because more demanding outcomes are often more valuable or are at least evaluated as such [14–17]. Effort is perceived as valuable in itself and may increase the value of an outcome [15]. This effect is universal and operates for various types of effort in the form of time, physical exertion, pain or money. Unrewarded efforts that do not lead to a desired goal—that is, efforts made in vain—demotivate action and are often devaluated retrospectively [15,18].

With this research we ask, what are the consequences of efforts in vain, which later appear not to serve any purpose. Learned helplessness is probably the most studied consequence of recurrent meaningless efforts in pursuit of an expected outcome. A prototypical study showed that individuals give up and fall into resignation or withdrawal when they see no relations between actions and their outcome [6]. Most of the research that has focused on this phenomenon has revolved around the issue of control, which is the core of learned helplessness. However, a lack of control over the expected causal links between actions and outcomes is also characteristic of meaningless situations. From a motivational point of view, learned helplessness manifests itself in inaction and passiveness. Accordingly, we predicted that meaningless situations that engage people's efforts may also demotivate individuals to act. For example, a parent who cannot clean up after his or her toddler, despite his or her efforts, probably sees no sense in engaging in further cleaning as it does not serve any purpose.

Noteworthy, not all pursuits that require effort but do not conclude in a desired goal are meaningless. Situations in which effort is made in vain but serves a purpose, e.g., fulfilling a long-term or a superordinate goal may not be perceived as meaningless. For example, a lost football game usually is not perceived as a meaningless effort, as each game may serve as a new experience and serve the goal of gaining skills that may allow attaining goals in the future. A person who has studied for an exam but failed it may see meaning in the act of acquiring more knowledge and thus be able to rationalize their efforts even though they were invested in vain.

Accordingly, the meaningfulness of an activity can be considered a subcategory of meaningfulness in general. Van Tilburg and Igou [9] showed that perceptions of meaningfulness of particular behavior are in part formed by the extent to which such a behavior: (1) is associated with a valuable goal; and (2) is instrumental in approaching this goal. Tasks that satisfy both of these criteria are, on average, perceived as meaningful (e.g., making a substantial donation to a charitable cause), whereas tasks that do not satisfy both criteria are, on average, perceived as meaningless.

### 1.3. Affective Consequences of Meaningless Effort

Two approaches have dominated the study of emotions and their consequences: (1) the valence-based approach, which generally focuses on positivity and negativity; and (2) the specific emotions approach, which relies on the appraisal theory of emotion. The latter theory posits that specific emotions of the same valence have distinct, idiosyncratic features related to their antecedents, phenomenology and outcomes [19,20]. It appears that distinguishing between seemingly similar emotions allows for the detection of subtle differences in behavioral consequences in similar situations [19]. This approach provides more detailed knowledge of specific behavioral consequences of experienced emotions. For example, client dissatisfaction manifests itself differently depending on whether the

client feels disappointment or regret. Disappointment has been found to be less likely to lead to client formal complaints than experienced regret [20].

Intuitively, situations of meaningless effort are not nice and pleasant experiences. Thus far, independent lines of research have studied affective consequences of effort in vain and, separately, of meaningless situations. However, the specific emotions that emerge in meaningless situations have not yet been explored. Previous studies regarding this matter have focused more broadly on negative affect. Research has shown that meaningless situations and failures to find meaning may enhance negative emotions (e.g., [11]). For example, it is known that novel, unexpected and discrepant circumstances resulting in surprise have a negative valence [21] and unsuccessful attempts to find meaning may evoke negative affect [22,23]. People generally react negatively to uncertainty (e.g., [24,25]) and discrepancy [26]. In our research, we expand previous research by exploring more specific emotions than relying on general valence only.

Relying on the appraisal theory of emotion, we focused specifically on disappointment, rather than on a general valence, evoked by meaningless situations. The literature suggests that surprising situations that do not lead to expected outcomes tend to lead to disappointment [27]. We think that two specific features of disappointment may play a relevant role in the effects of Sisyphean efforts on motivation to act further. First, disappointment is typically experienced when effort has been invested in vain and actions do not lead to the expected outcomes [7,28,29]. In particular, disappointment tends to arise in situations of limited self-agency, in which non-achievement of an outcome is attributed externally (e.g., to unfavorable circumstances [30,31]). Perceived agency is what distinguishes disappointment from regret which also arises in situations of effort invested in vain [31]. Second, disappointment is usually accompanied by frustration, powerlessness and perceived lack of control [7]. One likely consequence of disappointment is a tendency to do nothing, which often manifests itself in the forms of inaction, resignation and inertia [32]. For example, Martinez, Zeelenberg and Rijsman [33] found that disappointment decreases engagement in prosocial behaviors.

### 1.4. Research Goals

To the best of our knowledge, research to date has not focused on exploring the motivational effects of effort and meaninglessness as two compound predictors. Even if meaninglessness manipulations have required some effort on the part of participants in previous studies, effort has not been controlled and examined as an independent variable (e.g., [4]). Moreover, past research has been performed uniquely in laboratory settings with rather artificial tasks. In this research, we tested a model in which the meaninglessness of a behavior is moderated by the amount of effort invested in the pursuit of the goal and predicts the actor's motivation to engage in future conservation efforts.

Thus far, meaningless situations have been found to evoke negative affect. We expected that meaningless situations might evoke disappointment because both meaninglessness and disappointment are accompanied by powerlessness and violated expectations regarding the accomplishment of a goal. It is also known that not reaching a goal may result in disappointment. Accordingly, the interaction of meaninglessness and effort should increase disappointment. Disappointment, in turn, should negatively affect motivation. Hence, we proposed and tested a model in which meaninglessness, moderated by the level of effort investment, affects disappointment, which in turn relates to a decrease in motivation to engage in further conservation efforts.

However, historically much research on meaning and its consequences for motivation and behavior has neglected specific affective markers of this process, with only a handful of exceptions (e.g., curiosity and interest [34]). This is especially the case for negative affective states, which have been all but neglected. Indeed, while recent work by Maher et al. [35] discovered that meaninglessness and the (in)ability to find meaningful engagement are key characteristics of various negative affective states (including disappointment), studies that seek to model the role of specific emotions in the link from meaning to motivation remain

few. Our work addresses this issue by treating disappointment as key mediating variable in the link between meaning and behavioral intentions. In so doing, our work contributes to the understanding of what role specific emotions play when faced with meaningful versus meaningless situations. The specific affective consequences of such situations have not been studied and the literature does not address any specific emotions.

Furthermore, our objective is to test proposed models not only in hypothetical scenarios (Studies 1 and 2) but also in ecologically valid settings in the participants' homes (Study 3). In Study 3, we aimed to explore natural reactions. In our view, such an approach not only will contribute to the quality of our research, but will also create an opportunity to locate our findings in a broader context of conservation motivation.

## 2. Study 1

In Study 1, we tested whether an imagined scenario in which someone's effort related to recycling plastic bottles was wasted in a meaningless way by an experimenter, who puts everything to one bag, makes people experience disappointment. We also looked at whether such a situation would affect people's perceived sense in continuing an undertaken behavior. We expected the main effect of the meaningless condition on sense in continuing. We also hypothesized that the amount of effort necessary to complete the task would moderate the effect of the meaningless situation on disappointment and perceived sense in continuing. Additionally, disappointment should mediate the effect of the meaningfulness of the situation on the perceived sense in continuing. This mediation should be moderated by the amount of effort required by the task. We expected the mediating effect to arise solely with regards to disappointment, not other theoretically less relevant emotions.

### 2.1. Methods

#### 2.1.1. Participants

We recruited volunteers to the study via SONA recruitment system at the first author's university from a pool of psychology undergraduates studying at the university. Of the initial sample ($N = 273$), 20 (7.33%) participants failed to complete the entire procedure and 14 (5.13%) spent fewer than 10 s reading the scenario (before the study, we measured that 10 s is a minimum necessary to read the scenario). These participants were excluded from the final sample. Therefore, responses from 239 participants were included into the analyses. The age of participants ranged from 18 to 53 years ($M = 26.81$, $SD = 8.34$). The majority were women ($n = 205$, 85.8%). Women and men did not differ in average age, $F(1, 232) = 0.003$, $p = 0.956$, $\eta_p^2 = 0\%$, and perceived sense in continuing, $F(1, 232) = 2.15$, $p = 0.144$, $\eta_p^2 = 1\%$. Women were, however, more disappointed ($M = 3.00$, $SD = 1.26$) after reading the scenario than men ($M = 2.31$, $SD = 1.39$), $F(1, 233) = 7.41$, $p < 0.01$, $\eta_p^2 = 15\%$.

As this study was the first one testing the effect of meaningful effort, we had no ground to assume specific effect sizes. At the same time, previous research by Ariely et al. [4] found no significant difference between the control and the meaningless condition (two-tailed $p = 0.24$) with a sub-sample of $n = 69$. We thus assumed small effect sizes of $\eta^2 = 0.05$. For such effect, a sample size of at least $N = 201$ was determined, assuming power $(1 - \beta) = 0.90$, probability level $\alpha = 0.05$ and four experimental groups using G*Power software [36].

#### 2.1.2. Design and Procedure

Participants were randomly assigned to one of four conditions in a $2 \times 2$ design with the effort required by the task (low effort = 0 vs. high effort = 1), and the meaninglessness of the situation (meaningful = 1 vs. meaningless = 0) as between-subject conditions. In the low-effort condition (0), participants imagined that they put plastic bottles and screw caps into separate containers. In the high-effort condition (1), label bands also had to be torn off and sorted separately. Participants in the meaningful condition (1) imagined a situation in which, after recycling plastic bottles for seven days in athemiddle of a 14-day experiment, an assistant collected the bottles in separate bags. Participants in the meaningless condition

(0) imagined a situation in which recycled bottles were put jointly into one bag by the assistant.

Note that such a manipulation of meaninglessness is coherent with previously used manipulations in which a Lego robot constructed by a participant was destroyed by an experimenter [4] or pictures with marked tumor cells were deleted [5]. After reading the scenarios, participants were asked to what extent they would feel disappointment, sadness, frustration or anger in the given situation and whether they would see sense in continuing recycling bottles if they were in the given situation (for detailed instructions, see Appendices A.1 and A.2). Students received credit points in SONA system for the participation in the study. The Faculty Ethics Review Board from the first author's institution approved the experimental procedure. All analyses were performed using IBM SPSS Statistics version 26 software. Mediation and moderated mediation models were tested with the PROCESS macro [37].

### 2.1.3. Measures

*Emotions* were measured with straightforward questions about the extent to which participants felt them. Responses were given on a five-point scale from "very weakly" (1) to "very strongly" (5). The mean score for disappointment in the entire sample was $M = 2.90$, $SD = 1.29$. We also controlled for negative emotions, that is sadness, anger and frustration. These emotions were compared to disappointment in previous research (see [7]). As internal consistency of the three negative emotions was high (Cronbach's $\alpha = 0.82$), we collapsed them to create one variable. The mean score of these negative emotions was $M = 2.36$, $SD = 1.05$. The bivariate correlation between disappointment and negative emotions was $r = 0.76$, $p < 0.01$.

*Sense in continuing the task.* We measured sense in continuing with one question: "Please, indicate to what extent recycling bottles in the next week of the study would seem meaningful/meaningless to you?" Responses were given on a five-point scale from "meaningless" (1) to "meaningful" (5). The mean score for perceived sense in continuing was $M = 3.16$, $(SD = 1.51)$.

*Meaningfulness of the task.* To test the effectiveness of the manipulation of the meaninglessness, we used one question: "Was the behavior of the research assistant meaningful?" to which participants responded to using a five-point scale from "meaningless" (1) to "meaningful" (5).

### *2.2. Results*
### 2.2.1. Manipulation Check

The results of the analysis confirm that manipulation of meaninglessness was effective (see for manipulation check analysis in Appendix A.3).

### 2.2.2. Effect of Meaningless Effort on Perceived Sense in Continuing

A univariate analysis of variance with perceived sense in continuing recycling as a dependent variable yielded a significant main effect of the meaninglessness of the situation, $F(1, 237) = 41.41$, $p < 0.001$, $\eta_p^2 = 15\%$, with more sense perceived in continuing in the meaningful ($M = 3.71$, $SD = 1.27$) compared to meaningless condition ($M = 2.54$, $SD = 1.53$). There was no significant main effect of effort, $F(1, 237) = 0.54$, $p = 0.464$, $\eta_p^2 = 0\%$, and no interaction between the effort and the meaninglessness condition, $F(1, 237) = 1.25$, $p = 0.264$, $\eta_p^2 = 1\%$ (see Figure 1).

### 2.2.3. Moderated Mediation Test of Meaningless Effort on Sense in Continuing through Disappointment

Next, we tested whether the described mediation model depended on the amount of effort required by the task. To test the moderated mediation hypothesis, Model 7 of the PROCESS macro [37] with a bias-corrected bootstrapping procedure (10,000 samples) was used. The meaningfulness of the task had a significant effect on experienced disappoint-

ment ($\beta = -0.56$, $t = 10.37$, $p < 0.001$, $95\%CI$ $-0.67$, $-0.45$). The effect of the meaningfulness of the task on disappointment was significantly moderated by effort ($\beta = -0.12$, $t = -2.16$, $p = 0.032$, $95\%CI$ $-0.22$, $-0.01$). Disappointment was negatively related to perceived sense in continuing the conservation behavior ($\beta = -0.42$, $t = -6.18$, $p < 0.001$, $95\%CI$ $-0.55$, $-0.28$). Moreover, the bootstrapped moderated mediation effect for disappointment as a mediator and effort as a moderator was significant ($0.10$, $se = 0.05$, $95\%CI$ $0.01$, $0.20$).

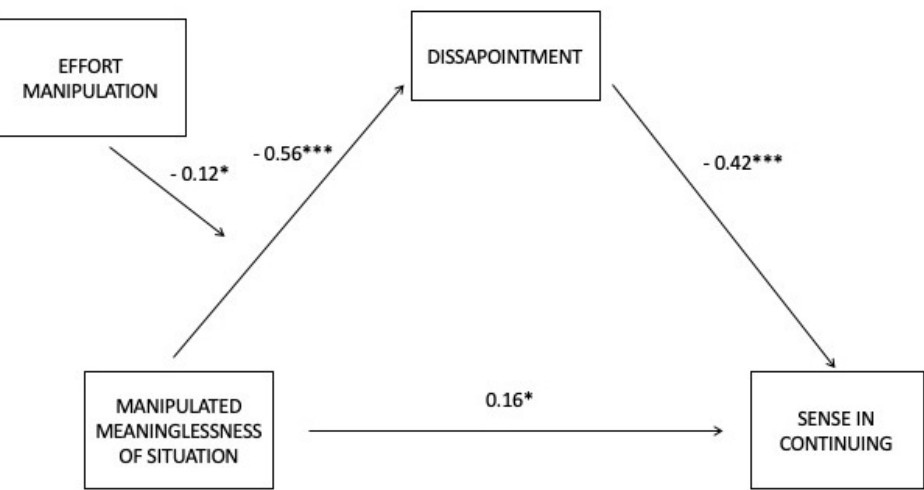

**Figure 1.** The standardized regression coefficients for the effect of meaningless situation moderated by effort required to complete the task on perceived sense in continuing recycling as mediated by disappointment (Study 1). Note that asterisks indicate significance levels; * $p < 0.05$, ** $p < 0.01$, *** $p < 0.001$.

Specific results show as expected that persons in the meaningless and the high-effort conditions experienced more disappointment than those in any other condition, and disappointment, in turn, affected perceived sense in continuing undertaken behavior (for detailed analysis of the interaction effect of meaninglessness and effort on disappointment, see Appendix A.4). Finally, we ran a sensitivity analysis to test whether the model holds after including general negative emotions. The results show that disappointment mediates the effect beyond and above other relevant emotions (see Appendix A.5 for detailed analyses).

*2.3. Discussion*

In Study 1, we found that participants exposed to a scenario in which the task was meaningless and required high-effort experienced disappointment. It appears that the mechanism of meaningless effort should be assigned to perceived disappointment. Data confirm the main effect of meaninglessness on disappointment and sense in continuing and the interaction of effort and meaninglessness on disappointment. We did not find the expected interaction of the effort and meaninglessness on sense in continuing, but effort was likely difficult to imagine in a scenario paradigm. Optionally, asking about sense in continuing focuses attention on cognitive aspect of the situation and not on its motivation. We thus performed one more study in this paradigm to expand these findings.

**3. Study 2**

In Study 2, we aimed to replicate findings from Study 1 using a different variable indicating motivation. Namely, we asked participants whether they would intend to continue a behavior for the next week. We used the same scenario as in Study 1, but, in the instruction describing the task in the effortful condition, we added the phrase "you tear off labels arduously" to emphasize that the task required some effort. Additionally,

we controlled for variables that might have affected the results or that might help in their understanding. Specifically, we measured participants' perceived meaningfulness, rationality and usefulness of the task and whether the task seemed difficult, time consuming and effortful. We also measured regret as an additional variable as previous research has suggested that disappointment and regret result from effort invested in vain, but their appearance depends whether a person has an agency over situation or not (e.g., [31]).

### 3.1. Methods

3.1.1. Participants

We recruited participants, psychology undergraduates, via SONA system at the first author's university. Taking part in Study 1 was an exclusion criterion for participation in Study 2 set automatically in the system. Of the initial sample ($N$ = 322), 39 (12.10%) spent fewer than 10 s reading the scenario. These participants were excluded from the final sample. Therefore, responses from 283 participants were included into the analyses. The age of participants ranged from 19 to 61 years ($M$ = 28.00, $SD$ = 8.73). The sample was more equivalent in terms of gender than in Study 1; 191 participants were female (68.0%) and 90 were male (32.0%).

A sample size of at least $N$ = 200 was determined based on a small effect size in the interaction analysis found in Studies 1 and 2. We assumed power $(1 - \beta)$ = 0.90, probability level $\alpha$ = 0.05 and small effect sizes ($\eta^2$ = 0.05).

3.1.2. Design and Procedure

The procedure in Study 2 replicated the procedure of Study 1. After reading the scenarios, participants were asked to what extent they would feel disappointment, sadness, frustration, anger and regret in the given situation and whether they would intend to continue recycling bottles if they were in the given situation. Then, they responded to additional manipulations checks and control questions. As in Study 1, students received credit points in SONA system for the participation in the study. The Faculty Ethics Review Board from the first author's institution approved the experimental procedure. All analyses were performed using IBM SPSS Statistics version 26 software. Mediation and moderated mediation models were tested with the PROCESS macro [37].

3.1.3. Measures

*Emotions* were measured with straightforward questions about the extent to which participants felt them. Responses were given on a five-point scale from "very weakly" (1) to "very strongly" (5). The mean score for disappointment in the entire sample was $M$ = 2.98, $SD$ = 1.43. As in Study 1, we controlled for negative emotions, that is sadness, anger and frustration. Additionally, we measured regret. An internal consistency of the three negative emotions was high (Cronbach's $\alpha$ = 0.84). The mean score of these negative emotions was $M$ = 2.73, ($SD$ = 1.16) and of regret was $M$ = 2.61, ($SD$ = 1.28). The bivariate correlation between disappointment and negative emotions was $r$ = 0.79, $p < 0.001$ and between disappointment and regret was $r$ = 0.77, $p < 0.001$.

*Intention to continue the behavior.* We measured intention with a question: "Please, indicate to what extent would you intend to recycle bottles in the next week of the study." Responses were given on a five-point scale from "definitely not" (1) to "definitely yes" (5). The mean score for the intention to continue was $M$ = 3.06, ($SD$ = 1.24).

*Meaningfulness of the situation.* We evaluated perceived meaningfulness of the assistant's behavior asking participants: "Was the behavior of the research assistant meaningful?" and we created an index of the meaningfulness of the task based on responses to three evaluation "Indicate to what extend the task was meaningful/useful/rational" to which participants responded to using a five-point scale from "meaningless" (1) to "meaningful" (5). An internal consistency of the three items was Cronbach's $\alpha$ = 0.90.

*Difficulty of the task.* Participants were asked to indicate to what extend the task was difficult, time-consuming, and demanding effort. Responses were given on a five-point

scale from "definitely not" (1) to "definitely yes" (5). We created an index of perceived difficulty, which had an internal consistency Cronbach's $\alpha = 0.80$.

*3.2. Results*
3.2.1. Manipulation Check

The results of the analysis confirmed that manipulation worked both when considered meaningfulness of the task and of an assistant. A univariate analysis of variance with the meaningfulness of an assistant's behavior as a dependent variable yielded a significant main effect of the meaninglessness manipulation, $F(1, 282) = 101.27$, $p < 0.001$, $\eta_p^2 = 27\%$. Participants in the meaningless condition ($M = 1.94$, $SD = 0.87$) perceived an assistant's behavior as less meaningful compared to the meaningful condition ($M = 3.19$, $SD = 1.14$). The main effect of effort was also significant, $F(1, 282) = 12.17$, $p < 0.001$, $\eta_p^2 = 6\%$. Assistant's behavior was perceived as more meaningful in the effortful ($M = 2.76$, $SD = 1.21$) compared to the low-effort condition ($M = 2.30$, $SD = 1.11$). The level of effort moderated the effect of the meaninglessness of the situation on perceived meaningfulness was marginally nonsignificant, $F(1, 282) = 3.76$, $p = 0.053$, $\eta_p^2 = 1\%$.

Additionally, we performed the ANOVA with the index of meaningfulness of the task. We observed a marginally nonsignificant main effect of the meaninglessness manipulation, $F(1, 282) = 2.78$, $p = 0.096$, $\eta_p^2 = 1\%$, but the means showed that participants in the meaningless condition ($M = 3.40$, $SD = 1.18$) perceived the task as less meaningful compared to the meaningful condition ($M = 3.64$, $SD = 0.97$). We also found the main effect of effort invested $F(1, 282) = 4.26$, $p = 0.040$, $\eta_p^2 = 2\%$. The task appeared more meaningful in the high effort ($M = 3.37$, $SD = 1.18$) than in the low-effort condition ($M = 3.65$, $SD = 1.00$). The effect of interaction was nonsignificant, $F(1, 282) = 0.91$, $p = 0.341$, $\eta_p^2 = 0\%$.

Although these results confirmed that manipulation worked, it is noteworthy that participants evaluated assistant's behavior as much less meaningful ($M = 1.94$, $SD = 0.87$) than they evaluated meaningfulness of the task ($M = 3.40$, $SD = 1.18$). This finding suggests that meaning of performed tasks was dependent not only on external situation of the manipulation, but also on subjective perception of the situation by participants. The analysis also yielded a nonsignificant main effect of effort manipulation. For more manipulation check analyses, refer to Appendix B.1.

3.2.2. Effect of Meaningless Effort on Intention to Continue Recycling

A univariate analysis of variance yielded a significant main effect of the meaninglessness of the situation $F(1, 282) = 15.95$, $p < 0.001$, $\eta_p^2 = 5\%$, with more sense perceived in continuing in the meaningful ($M = 3.36$, $SD = 1.13$) compared to meaningless condition ($M = 2.77$, $SD = 1.27$). There was no significant main effect of effort, $F(1, 282) = 2.58$, $p = 0.109$, $\eta_p^2 = 1\%$, and no interaction between the effort and the meaninglessness condition, $F(1, 282) = 0.12$, $p = 0.725$, $\eta_p^2 = 0\%$.

3.2.3. Moderated Mediation Test of Meaningless Effort on on Intention to Continue Recycling through Disappointment

Finally, we test the moderated mediation hypothesis with Model 7 of the PROCESS macro [37] with a bias-corrected bootstrapping procedure (10,000 samples). The meaningfulness (1 = meaningful, 0 = meaningless) of the task had a significant effect on experienced disappointment ($\beta = -0.19$, $t = -4.07$, $p < 0.001$, $95\%CI$ $-0.29$, $-0.10$). The effect of the meaningfulness of the task on disappointment was significantly moderated by effort ($\beta = -0.10$, $t = -2.07$, $p = 0.040$, $95\%CI = -0.19$, $-0.005$). Disappointment was negatively related to intention to continue the conservation behavior ($\beta = -0.28$, $t = -4.18$, $p < 0.001$, $95\%CI$ $-0.42$, $-0.15$). Finally, the bootstrapped moderated mediation effect for disappointment as a mediator and effort as a moderator was significant (0.06, $se = 0.03$, $95\%CI$ 0.003, 0.13; see Figure 2). For detailed analysis of the interaction effect of meaninglessness and effort on disappointment, see Appendix B.2. As in Study 1, we performed a sensitivity analysis of the model, which showed that disappointment mediated the effect beyond

regret, but when other negative emotions (sadness, anger and frustration) were introduced the mediating effect of disappointment disappeared (see Appendix B.3).

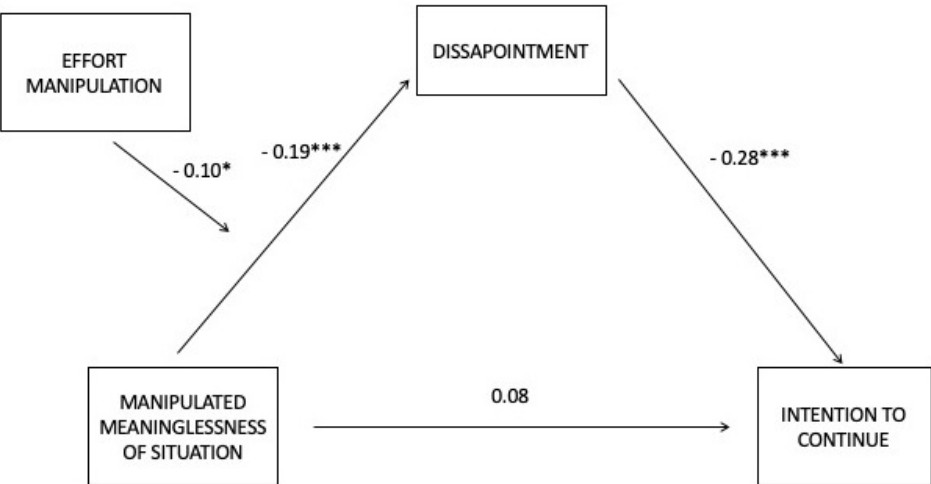

**Figure 2.** The standardized regression coefficients for the effect of meaningless situation moderated by effort required to complete the task on intention to continue recycling as mediated by disappointment (Study 2). Note that asterisks indicate significance levels; * $p < 0.05$, ** $p < 0.01$, *** $p < 0.001$.

*3.3. Discussion*

The manipulations affected perceived meaningfulness of the assistant's behavior, but not the meaningfulness of the task. Perhaps, people assign meaning to the tasks they perform depending on different indicators. Meaninglessness affected intention to continue behavior. Consistently as in Study 1 with sense in continuing the task, we found no main effect of effort and no interaction. Probably, the effect of imagined effort in scenario was not strong enough to affect hypothetical intentions to continue the task.

Finally, we confirmed the role of disappointment as a mediator of the effect. Nonetheless, in this study, a highly reliable measure of three emotions, highly correlated with disappointment, explained more variance in intention when entered as a second mediator to the moderated mediation model. In the model in which we entered regret, disappointment affected intention to continue behavior beyond and above that similar emotion.

It is noteworthy that one of the limitations of Studies 1 and 2 was that participants responded to a scenario and were asked to imagine emotions. Although many studies in psychology are conducted in such paradigms, we decided to replicate these findings in a more ecologically valid situation.

## 4. Study 3

In Study 3, we explored the effect of the same experimental conditions of meaninglessness and effort required by a task but in a longitudinal field experiment in a real-life setting. Moreover, our objective was to validate findings from Studies 1 and 2 to see whether experiencing meaninglessness affects performing an actual behavior and intention to engage in future conservation behaviors. Finally, we found it important to confirm the mediating role of disappointment between the meaninglessness of the situation, intentions and actual behavior.

### *4.1. Methods*

### 4.1.1. Study Design

We performed a 2 × 2 × 2 field experiment with random allocation to the effort required by the task (low effort = 0 vs. high effort = 1 condition) and meaninglessness (meaningless = 0 vs. meaningful = 1 condition) as between-subject factors and time of measurement ($t_0 - t_2$) as a within-subject factor. Disappointment, a mean number of bottles recycled per day and an intention to engage in various conservation behaviors were dependent variables. To explore the role of disappointment, we performed a moderated mediation analysis, testing the same model as in Studies 1 and 2. The perceived meaningfulness of the task was a predictor of intention to engage in other conservation behaviors at $t_2$, and experienced disappointment was tested as a mediator. Effort required by the task was a moderator on the path from the perceived meaningfulness and disappointment. All analyses were performed using IBM SPSS Statistics version 26 software. Mediation and moderated mediation models were tested with the PROCESS macro [37].

### 4.1.2. Participants

We recruited participants via the authors' university website, social media profiles and using a snowball procedure, that is by asking participants at the time of enrolment to recommend the study to their friends and colleagues. We collected data in the urban area of Wroclaw and Walbrzych, cities in southwest Poland. A sample size of at least $N = 200$ was determined based on a small effect size in the interaction analysis found in Studies 1 and 2. We assumed power $(1 - \beta) = 0.90$, probability level $\alpha = 0.05$ and small effect sizes ($\eta^2 = 0.05$), at least two measurement and correlation between repeated measurement $r = 0.45$.

Initially, 345 individuals volunteered for the study and completed the base measurement ($t_0$). Of these, 286 (82%) completed the entire procedure ($t_0 - t_2$). The participants' mean age was $M = 40.18$ ($SD = 13.61$), ranging from 18 to 87 years, and 54.9% ($n = 157$) were female. Most participants had a university degree (53.1%) or had completed high school (32.9%); 10.5% reported middle school and 1% elementary school as their highest level of educational attainment. On average, participants evaluated their perceived socio-economic status as neither higher nor lower than that of an average family in Poland, $M = 3.08$ ($SD = 0.87$), when responding to a five-point scale ranging from "much below the status of an average family" (1) to "much above the status of an average family" (5).

Participants who dropped out ($n = 59$, 17.1%) and those who completed the entire procedure did not differ in gender, $\chi^2(1, N = 344) = 0.078$, $p = 0.780$, $\phi = -0.02$; education, U-test = $-1.69$, $p = 0.091$; or perceived socioeconomic status, $F(1, 322) = 1.06$, $p = 0.305$, $\eta_p^2 = 2\%$. However, participants who dropped out of the study were significantly younger ($M = 34.70$, $SD = 11.54$) than those who completed the study ($M = 40.18$, $SD = 13.61$), $F(1, 328) = 6.80$, $p < 0.05$, $\eta_p^2 = 2\%$. Observations with missing values were deleted list-wise during the analyses. Little's MCAR (after including disappointment, negative emotions, mean bottles recycled per day, intention measured at three points and demographic variables) confirmed that missing values appeared at random, $\chi^2(1, N = 344) = 45.81$, $p = 0.104$.

### 4.1.3. Procedure

During online registration, participants received information about the study procedures and gave informed consent. They were informed that the aim of the study was to evaluate the functionality of a rotary recycle bin. To make this cover story more realistic participants were asked about the functionality of the bins at the end of the study. Next, they completed a baseline questionnaire ($t_0$), including sociodemographic variables as well as other scales administered as part of a larger research project on conservation behaviors. At this stage, participants also provided their e-mail addresses and phone numbers.

After the questionnaire completion, participants were contacted to schedule an appointment with a research assistant. Upon each participant's arrival, an assistant provided a recycling bin with three separate containers labeled "bottles," "screwcaps" and optionally "label bands" if a participant was assigned to high-effort condition. Participants also received fourteen 0.33 L (0.11 oz) bottles of water. In the low-effort condition, assistants instructed participants to put plastic bags and screw caps into separate containers. In the high-effort condition, they were also instructed to tear off label bands so that they can be recycled.

After approximately one week ($M = 6.93$ days, $SD = 0.54$, ranging from 5 to 9 days), an assistant collected and counted empty bottles, screw caps and label bands if applicable. In the meaningful condition (1), sorted items were collected to separate plastic bags. In the meaningless condition (0), all items were put into one bag. The sorting of the items was performed in front of a participant. The assistant was additionally instructed to describe their actions verbally and naturally while doing them, i.e., "Right, I will now put everything into one/separate bags in line with the instructions I received." Then, participants were asked to complete questionnaire forms ($t_1$). They received a set of 14 bottles with the same instructions as at the first meeting. That is, participants remained in the low-effort or the high-effort condition as assigned at the beginning.

Approximately a week later ($M = 7.03$ days, $SD = 0.78$, ranging from 5 to 9 days), an assistant visited each participant and repeated the procedure from the week before. Additionally, participants evaluated the functionality of the rotary bin to be consistent with the cover story. At the last meeting, as compensation for their effort, each participant received a vacuum bottle worth ca. €15 (£10) and more details about the research objective. For more details on the procedure and instructions see Appendixs C.1 and C.2). The procedure of the study was approved by the Faculty Ethics Review Board.

### 4.1.4. Measures

*Emotions* were measured only at $t_2$ with straightforward questions ("To what extend do you feel?"), as in Study 1, and responses were given on a five-point scale from "very weakly" (1) to "very strongly" (5). The mean score for disappointment in the entire sample was $M = 1.64$, $SD = 0.98$. Negative emotions were measured with the same emotions as in Study 1: anger, frustration and sadness. The mean score for negative emotions was $M = 1.68$, $SD = 0.83$, Cronbach's $\alpha = 0.77$. In this study, we also measured positive emotions, as they could serve as a comparison standard for disappointment. For this, we used the International positive affect schedule short-form (I-PANAS-SF) by Thompson [38] consisting of five items: active, determined, attentive, inspired and alert. The mean score for positive emotions was $M = 2.91$, $SD = 0.77$, Cronbach's $\alpha = 0.74$. The bivariate correlation between disappointment and negative emotions was $r = 0.58$, $p < 0.001$ and disappointment and positive emotions were not correlated $r = 0.02$, $p = 0.734$.

*Conservation behavior* was measured in an objective way at two time points approximately one week ($t_1$) and two weeks ($t_2$) after the first visit of a research assistant in a household by counting the number of recycled bottles found by a research assistant in a recycle bin minus the number of bottles that participants declared they did not recycle by themselves. On average, participants recycled $M_{t1} = 10.85$ ($SD_{t1} = 3.31$), $Med_{t1} = 12.00$ bottles in the first week and $M_{t2} = 11.15$ ($SD_{t2} = 3.29$), $Med_{t2} = 12.00$ in the second week. Periods between two measurements were on average seven days, but it was longer or shorter for some individuals because of pragmatical reasons (e.g., unexpected rescheduling of an appointment with an assistant). Therefore, we used the number of empty bottles recollected at a measurement time minus the number of bottles participants declared they did not recycle by themselves divided by a number of days over which a participant took part in each period as our DV. On average, participants recycled $M_{t1} = 1.58$ ($SD_{t1} = 0.50$) bottles per day in the first week and $M_{t2} = 1.60$ ($SD_{t2} = 0.50$) in the second week. The variables at two points of measurement were correlated at $r = 0.53$, $p < 0.001$.

*Intention* to engage in future conservation behaviors was measured with three questions taken from the Polish version (see [39]) of the General Environmental Behavior Scale [40]. The items were formulated as follows: "In the future after a picnic I'm going to leave a place as clean as I found it; In the future, I intend to take a shower rather than a bath in a bathtub; In the future, I intend to buy energy-saving appliances and audio/video devices." Responses ranged from "strongly disagree" (1) to "strongly agree" (5). Internal consistency of the three items was acceptable, taking into account that Cronbach's coefficient is dependent on the number of items, $\alpha_{t0} = 0.65$, $\alpha_{t1} = 0.63$, $\alpha_{t2} = 0.70$, e.g., [41]. The mean score for intention in the entire sample at three measurements was as follows: $M_{t0} = 4.43$, $SD_{t0} = 0.59$, $M_{t1} = 4.45$, $SD_{t1} = 0.68$, $M_{t2} = 4.47$, $SD_{t2} = 0.73$.

*4.2. Results*

4.2.1. Pilot Study and Manipulation Check

Before the main study, we performed a pilot study in which we found that effortful condition required significantly more time to recycle bottles and that it was perceived as more effortful, difficult and laborious. In the manipulation check, we found that persons in the meaningless condition searched for meaning more than those in the meaningful condition. We found, however, no effect of the manipulation on the perceived meaningfulness of the task (see Appendix C.3 for details on manipulation check and pilot study).

4.2.2. Effect of Meaningless Effort on Intention to Engage in Conservation Behaviors

We found no effect of manipulations on recycling, therefore we focus on behavioral intention to act pro-environmentally (see details in Appendix C.5). Prior to the main analysis, we measured whether experimental groups differed in the level of intention to engage in conservation behaviors at $t_0$. A univariate analysis of variance showed no main effects of the meaninglessness, $F(1, 281) = 0.59$, $p = 0.445$, $\eta_p^2 = 0\%$ or effort, $F(1, 281) = 1.94$, $p = 0.165$, $\eta_p^2 = 1\%$, or an interaction of the experimental conditions, $F(1, 281) = 2.58$, $p = 0.109$, $\eta_p^2 = 1\%$, which confirms the effectiveness of random allocation to experimental groups.

A mixed-effects analysis of variance yielded no significant main effect of the time of measurement, $F(2, 458) = 0.29$, $p = 0.752$, $\eta_p^2 = 0\%$, and meaninglessness of the situation, $F(1, 228) = 1.39$, $p = 0.213$, $\eta_p^2 = 1\%$. We found, however, a main effect of effort, $F(1, 228) = 4.28$, $p < 0.05$, $\eta_p^2 = 2\%$, with participants in the low-effort condition having stronger intentions to engage in conservation behaviors ($M = 4.53$, $SD = 0.70$) than participants in the high-effort condition ($M = 4.40$, $SD = 0.78$). The analysis also yielded the expected interaction of effort invested and meaninglessness of a behavior, $F(1, 228) = 7.25$, $p < 0.01$, $\eta_p^2 = 3\%$, with participants in the meaningful and less effort condition being more motivated to engage in future conservation behaviors than those in all other conditions. Means associated with the interaction are shown in Figure 3. We found no other two-way interactions between the time of measurement × the meaninglessness of the situation, $F(2, 456) = 0.52$, $p = 0.594$, $\eta_p^2 = 0\%$, or the time of measurement × the effort required by the task, $F(2, 456) = 0.55$, $p = 0.576$, $\eta_p^2 = 0\%$. The three-way interaction of the time of measurement, meaninglessness and effort was also nonsignificant, $F(2, 456) = 1.38$, $p = 0.254$, $\eta_p^2 = 1\%$.

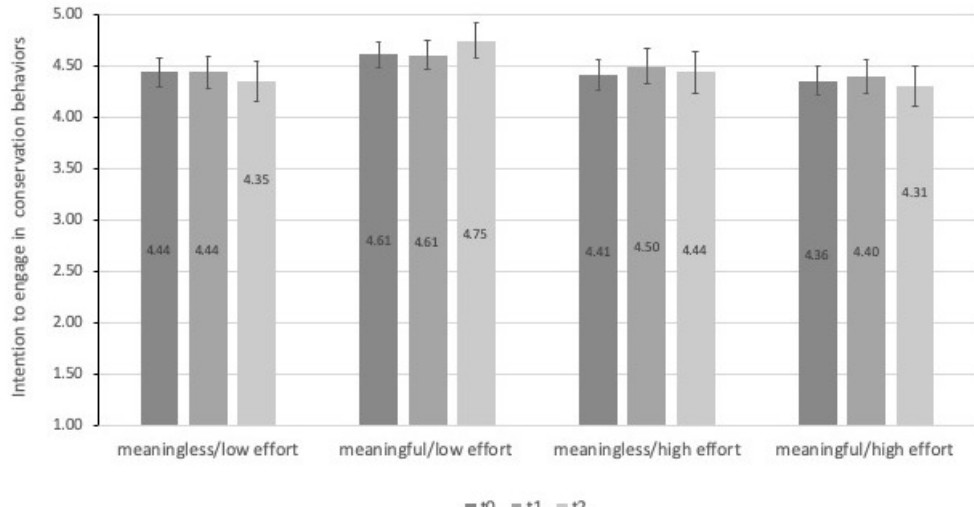

**Figure 3.** Effects of the meaninglessness of the situation and effort required to complete task on intention to engage in other conservation behaviors measured at three time points (Study 3). Note that vertical bars indicate 95% confidence intervals.

### 4.2.3. Moderated Mediation of Meaningless Effort on Intention to Engage in Conservation Behaviors through Disappointment

Next, we tested whether the moderated mediation model corroborated in Studies 1 and 2 was replicated in Study 3. We expected, again, that effort would moderate the relation between the perceived meaningfulness of the task and disappointment. Disappointment, in turn, should decrease people's intention to engage in future conservation behaviors. To test such a model, we performed a moderated mediation Model 7 of the PROCESS macro [37] with a bias-corrected bootstrapping procedure (10,000 samples). The analysis yielded a significant model of moderated mediation (0.06, $se$ = 0.04, 95%$CI$ 0.001, 0.15).

The perceived meaningfulness of the task had a significant effect on experienced disappointment ($\beta$ = −0.17, $t$ = −2.56, $p$ < 0.05, 95%$CI$ −0.30, −0.04). The interaction of the meaningfulness and difficulty also had an effect on disappointment ($\beta$ = −0.21, $t$ = −3.07, $p$ < 0.01, 95%$CI$ −0.34, −0.07). For details on the nature of the interaction, see Appendix C.4. Disappointment was negatively related to willingness to engage in future conservation behaviors at $t_2$ ($\beta$ = −0.14, $t$ = −1.99, $p$ < 0.05, 95%$CI$ −0.27, −0.001) when controlled for the intention at $t_0$. The intention at $t_0$ did not have a significant effect on disappointment ($\beta$ = −0.05, $t$ = −0.84, $p$ = 0.394, 95%$CI$ −0.18, 0.07), but quite intuitively it was related with the intention at $t_2$ ($\beta$ = 0.34, $t$ = 5.30, $p$ < 0.001, 95%$CI$ 0.22, 0.47) (for details, see Figure 4).

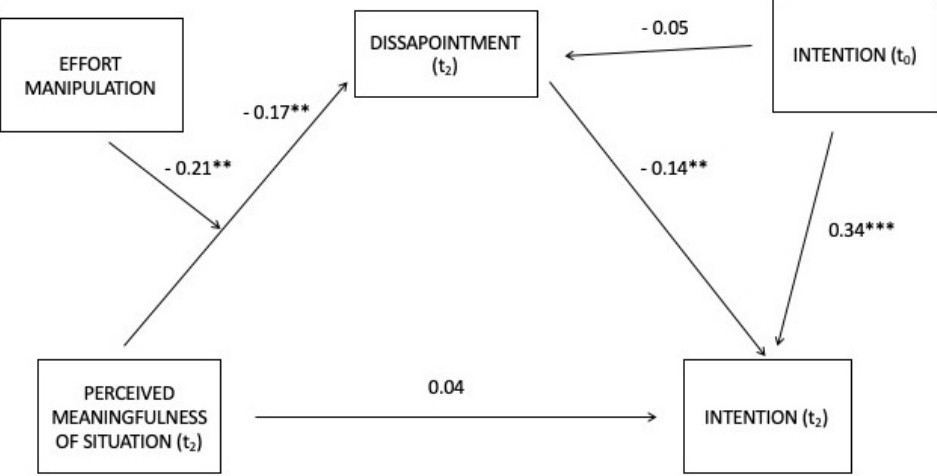

**Figure 4.** The standardized regression coefficients for the relationship between perceived meaningfulness of the situation moderated by effort required to complete the on the intention to engage in conservation behaviors as mediated by experienced disappointment (Study 3). Note that asterisks indicate significance levels; * $p < 0.05$, ** $p < 0.01$, *** $p < 0.001$.

Consistent with the findings of Studies 1 and 2, specific results show that persons perceiving the task as less meaningful who were in the difficult condition experienced more disappointment, and disappointment, in turn, affected their intention to engage in additional conservation behavior. We ran a sensitivity analysis entering general negative emotions as the second mediator to the model. The analysis showed that the effect of disappointment on intention when including negative emotions was marginally nonsignificant. The effect of disappointment on intention remained significant when including positive emotions (see Appendix C.6 for details).

*4.3. Discussion*

In sum, in Study 3, we found the effect of Sisyphean effort, that is the effect of interaction of meaninglessness and effort on intention to engage in other conservation behaviors such as buying energy-saving appliances in the future or taking more ecological shower rather than a bath. We found no effect of used manipulations on the number of bottles recycled during the study. Moreover, the manipulation of meaninglessness did not affect disappointment. However, we found the expected main effect of perceived meaningfulness of the task on disappointment and disappointment mediated the effect of the perceived sense in the given task and effort on intention to engage in further conservation behaviors.

**5. General Discussion**

The aim of the present studies was to test the compound effect of the meaninglessness of a task and the effort required for its realization on affect and on declines in motivation to engage in conservation actions. In all studies, we confirmed that Sisyphean efforts lead to demotivation through disappointment. We found that the meaninglessness of a situation resulted in a decrease in the level of perceived sense in continued efforts to recycle in the near future (Study 1), an intention to continue recycling (Study 2) and intention to engage in other conservation behaviors (Study 3). We did not find a direct moderation of effort on indices of motivation in Studies 1 and 2, but we found the effect in Study 3. Specifically, in Study 3, we found that intention to engage in other conservation behaviors is decreased in all conditions, but the condition in which the task was meaningful and required less effort. Sisyphean efforts did not affect the number of bottles recycled. However, this could be attributed to the measure of behavior that was used in the present study rather than an

actual lack of effect. Our participants were highly engaged in the study procedures and many of them recycled almost all the bottles they received.

Disappointment was a consistently significant mediator of the effect examined in the three studies. In line with our predictions, and in accordance with the appraisal of emotion approach, disappointment (rather than other emotions) played a mediating role between meaningless effort and indices of motivation in Studies 1 and 3. In Study 2, the measure of negative emotions explained more variance in intention than disappointment, yielding disappointment nonsignificant. In the same study disappointment was not affected by regret entered to the model and disappointment explained decline in motivation beyond and above regret. In general, as previously suggested by Zeelenberg et al. [31], the distinction between disappointment and other emotions was important in understanding the motivational processes and specific consequences of experienced emotions.

Some caution is warranted when interpreting the mediations by disappointment. Negative emotions similar to disappointment such as frustration or anger share much variance with disappointment, which made it difficult to distinguish its individual effect in models that contained them simultaneously in Studies 2 (Appendix B.3) and 3 (Appendix C.6). In Study 3, we measured positive emotions, which apparently did not play a role in the model (Appendix C.6). Meaningful effort could likely have given rise to specific positive emotions such as satisfaction or pride that we did not measure. Future research may examine further the role of specific negative and positive emotions in meaning-regulation and we hope that our work provides a basis for doing so.

It appears particularly interesting that meaningless effort affected intention to engage in other conservation behaviors. Although these results warrant replication with a broader scope of behaviors, our findings suggest that experienced disappointment may carry over onto other behaviors that serve the same goal (in this case, protection of the environment). Our findings may explain one of the mechanisms behind observed negative spillover effects, or situations in which engagement in one behavior leads to resignation from the intention to engage in other behaviors of the same category [42,43].

### 5.1. Imagined Scenario and Real-Life Setting

Previous research has found that people make different decisions in studies of imagined scenarios than in real-life studies [44,45]. We did not find any fundamental differences between the results of the three present studies, as both confirmed the role of disappointment, but some differences between these results should be discussed in the broader research context. The observed effect sizes were much higher in the Internet study than in the real-life setting. This is not surprising, as field experiments trade off internal validity for external validity. A whole spectrum of variables that are impossible to control in natural setting affects variance in the variables observed in field studies. Participants' reported emotions regarding imagined scenarios were more intense than actually experienced. The mean level of disappointment was lower in Study 3 than in Studies 1 and 2. This is in line with the findings that imagined and actually experienced emotions differ in intensity in the same direction as observed in the present studies [46]. However, the same specific emotion consistently played a role in the examined phenomenon in the three present studies.

### 5.2. Limitations

A few limitations are worth mentioning when interpreting our results. A principal limitation of Studies 1 and 2 was that they were based on an imagined scenario, but this limitation has been resolved through replication in real-life settings. Another limitation of the present research was that we focused on one specific behavior in both studies. Such an approach limits the generalizability of the findings. However, the advantage of comparing one behavior across three studies was that we managed to replicate the main results.

The manipulation of meaninglessness that worked in the hypothetical scenario of Studies 1 and 2 did not work for disappointment in the real-life setting of Study 3, although it worked for intention. It could have failed for a variety of reasons. For one, participants

could have perceived their behaviors as subjectively meaningful. Some individuals might have thought that their actions were part of the purposeful procedure that was unclear to them, but the fact that they are following the procedure by doing a specific task made sense to them. A research assistant clearly stated that, according to the procedure, he or she would put everything into one bag. The results of the manipulation check in Study 2 speak in favor of such an interpretation (see Appendix B.1).

Namely, perceived meaningfulness of an assistant's behavior depended on the manipulated conditions, in line with our predictions, but perceived meaningfulness of the task did not. Noteworthy, manipulations based on the destruction of performed work have been used before by Ariely et al. [4]. However, Ariely et al. [4] did not use a manipulation check to verify the meaninglessness manipulation. Interestingly, in a study by Chandler and Kapelner [5], this manipulation check worked only partially. Similar to the participants in our studies, the participants in Chandler and Kapelner's study did not perceive the meaningless condition as more meaningless than the participants in the control condition. This result suggests that evaluations of meaningfulness and meaninglessness may depend upon context, the characteristics of an individual and the relationship between an individual's goals and their actions [9,47].

The use of plastic bottles could have been perceived by participants as un-ecological behavior. Nonetheless, drinking tap water is not very common in Poland, and drinking from plastic bottles has become more and more popular with each year. Moreover, the cover story presented to participants stated that the aim of the study was to evaluate the functionality of a recycling bin. We also asked participants whether they perceived their task as related to environmental protection and controlled for their responses in a moderated mediation model. The results of the moderated mediation when controlling for perceived environmental protection of the task remain the same. Finally, in Study 2, the presence of an experimenter could have masked the effects of some variables. However, we did not find any significant differences in the numbers of bottles recycled, intentions or disappointment of participants between experimenters. Moreover, the presence of an experimenter helped us to control the study procedure, as it is a challenge in field experiments.

One may argue that the effect sizes of the discovered results are not strong enough to have a sufficient impact on real-life behavior. However, we believe that the significance of these findings relies on the scale. That is, recycling is relevant for many people, it is performed frequently, and it attracts media attention [1–3]. Thus, even the slight negative effects of wasted conservation efforts might have a broader impact by spreading negative emotions to other people through social media and media coverage or by being carried over to other conservation domains, consistent with work on 'nudging' to encourage particular behaviors. Undoubtedly, these implications need further verification in real-life settings. Our results do not provide support that the differences in behavioral intentions could translate to real conservation behavior. Future studies should test for possible spillover effects using longitudinal designs.

## 6. Conclusions

The three present studies allowed us to observe the affective and motivational consequences of meaningless effort both in imagined scenarios and in a real setting. We tested our hypothesis in the context of recycling, but future research should replicate our findings in other settings, such as contexts involving energy saving or pro-environmental protesting. Moreover, we focused on meaningful, effortful tasks that later became meaningless. The consequences of effortful engagement in behaviors that are meaningless to begin with require further exploration.

Altogether, we found the expected interaction effect of effort and meaninglessness on disappointment and intention to engage in other conservation behaviors. The meaninglessness of the given task affected perceived sense in continuing and intention to continue recycling. In three studies, we consistently confirmed the proposed moderated mediation

models in which disappointment acted as a mediator and effort acted as a moderator. Our research serves to enrich the literature on conservation motivation, showing that Sisyphean efforts lead to the decrease in motivation because of an affective component. We hope that our results will inspire exploration of the role of affect in promoting conservation behaviors. This line of work warrants more scholarly attention, as the findings in other domains, for example, decision making or health, have shown that emotions, such as regret or guilt, lead to behavioral change and spillover effects (see, e.g., [33,48]). Our research could serve as a stepping stone for studies investigating these processes in real-life settings.

People's conservation behaviors are key to sustainable development and climate change mitigation. Many solutions to environmental problems require changes in human behaviors [49,50]. However, some behaviors have already been adopted and vastly performed by people. Recycling is an example, as multiple studies have shown that the likelihood of engaging in its various forms is as high as 90% (see, e.g., [51,52]). Accordingly, scholars argue that little space remains for change in these popular behaviors, and, in such cases, policy makers should not focus on the promotion of behaviors but on their maintenance instead (see, e.g., [53]). We hope that our results will inspire interventions targeted at behavior maintenance by emphasizing the meaningfulness of people's sustainable activities and by showing that conservation efforts lead to purposeful goals.

**Author Contributions:** Conceptualization, K.B., K.C., D.D. and W.V.T.; methodology, K.B., K.C., D.D. and W.V.T.; formal analysis, K.B.; investigation, K.B.; data curation, K.B. and K.C.; writing—original draft preparation, K.B. and K.C.; writing—review and editing, K.B., K.C., D.D. and W.V.T.; project administration, K.B.; and funding acquisition, K.B. and K.C. All authors have read and agreed to the published version of the manuscript.

**Funding:** This research was financially supported by grant 2014/13/D/HS6/01423 from the National Science Center, received by Katarzyna Byrka, and by grant 2015/16/S/HS6/00254 from the National Science Center, received by Katarzyna Cantarero. Open access of this article was financed by the Ministry of Science and Higher Education in Poland under the 2019–2022 program "Regional Initiative of Excellence", project number 012/RID/2018/19.

**Institutional Review Board Statement:** The project was conducted according to the guidelines of the Declaration of Helsinki, and approved by the Institutional Review Board of the Faculty of Social Psychology (date of approval: 6 October 2016).

**Informed Consent Statement:** Informed consent was obtained from all subjects involved in the study.

**Data Availability Statement:** The datasets are available upon request or can be accessed via the link: https://osf.io/2kbhx/?view_only=290f7eef4dd54f4bb61dc02f4154989d, accessed on 2 May 2020.

**Acknowledgments:** We are grateful to Magdalena Dubas, Aneta Bartczak, Agata Morag and Anna Kaminska who helped with data collection. We thank Agata Gasiorowska for her insightful comments on the early daft of the manuscript.

**Conflicts of Interest:** The authors declare no conflict of interest.

## Appendix A. Study 1

The following are presented in this Appendix: (Appendix A.1) instructions translated into English; (Appendix A.2) original instructions in Polish; (Appendix A.3) manipulation check; (Appendix A.4) effect of meaningless effort on disappointment; and (Appendix A.5) sensitivity analysis.

*Appendix A.1. Instructions Translated into English*

**Group: meaningful/low effort**

Imagine the following situation: you are participating in a study exploring segregation of plastic bottles and assessing the functionality of waste bins. For one week, in your home you segregate empty bottles of mineral water you have received as part of the study—you

crush the bottle and put it and a cap into separate chambers of a waste bin. After a week, the person conducting the study comes and puts everything into separate bags, saying "I'm throwing everything into separate bags, in line with the instructions I received."

### Group: meaningless/low effort

Imagine the following situation: you are participating in a study exploring segregation of plastic bottles and assessing the functionality of waste bins. For one week, in your home you segregate empty bottles of mineral water you have received as part of the study—you crush the bottle and put it and a cap into separate chambers of a waste bin. After a week, the person conducting the study comes and puts everything into one bag, saying "I'm throwing everything into one bag, in line with the instructions I received."

### Group: meaningful/high effort

Imagine the following situation: you are participating in a study exploring segregation of plastic bottles and assessing the functionality of waste bins. For one week, in your home you segregate empty bottles of mineral water you have received as part of the study—you tear off the labels, crush them and put the bottle and cap, and labels into separate chambers of a waste bin. After a week, the person conducting the study comes and puts everything into separate bags, saying "I'm throwing everything into separate bags, in line with the instructions I received."

### Group: meaningless/high effort

Imagine the following situation: you are participating in a study exploring segregation of plastic bottles and assessing the functionality of waste bins. For one week, in your home you segregate empty bottles of mineral water you have received as part of the study—you tear off the labels, crush them and put the bottle and cap, and labels into separate chambers of a waste bin. After a week, the person conducting the study comes and puts everything into one bag, saying "I'm throwing everything into one bag, in line with the instructions I received."

*Appendix A.2. Original Procedure in Polish*

### Sens/mały wysiłek

Wyobraź sobie następującą sytuację. Bierzesz udział w badaniu dotyczącym segregowania butelek plastikowych i oceny funkcjonalności pojemników na odpady. Przez tydzień w domu segregujesz otrzymane w ramach badania butelki po wodzie mineralnej—zgniatasz i wyrzucasz butelkę i nakrętkę do osobnych komór pojemnika na odpady. Po tygodniu przychodzi osoba przeprowadzająca badanie i wrzuca wszystko do jednego worka, mówiąc: "Wrzucam wszystko do osobnych worków, zgodnie z procedurą"

### Bezsens/mały wysiłek

Wyobraź sobie następującą sytuację. Bierzesz udział w badaniu dotyczącym segregowania butelek plastikowych i oceny funkcjonalności pojemników na odpady. Przez tydzień w domu segregujesz otrzymane w ramach badania butelki po wodzie mineralnej—zgniatasz i wyrzucasz butelkę i nakrętkę do osobnych komór pojemnika na odpady. Po tygodniu przychodzi osoba przeprowadzająca badanie i wrzuca wszystko do jednego worka, mówiąc: "Wrzucam wszystko do jednego worka, zgodnie z procedurą"

### Sens/duży wysiłek

Wyobraź sobie następującą sytuację. Bierzesz udział w badaniu dotyczącym segregowania butelek plastikowych i oceny funkcjonalności pojemników na odpady. Przez tydzień w domu segregujesz otrzymane w ramach badania butelki po wodzie mineralnej—zgniatasz je, zdzierasz wszystkie etykiety i wyrzucasz butelkę, etykiety i nakrętkę do

osobnych komór pojemnika na odpady. Po tygodniu przychodzi osoba przeprowadzająca badanie i wrzuca wszystko do jednego worka, mówiąc: "Wrzucam wszystko do osobnych worków, zgodnie z procedurą"

**Bezsens/duży wysiłek**

Wyobraź sobie następującą sytuację. Bierzesz udział w badaniu dotyczącym segregowania butelek plastikowych i oceny funkcjonalności pojemników na odpady. Przez tydzień w domu segregujesz otrzymane w ramach badania butelki po wodzie mineralnej—zgniatasz je, zdzierasz wszystkie etykiety i wyrzucasz butelkę, etykiety i nakrętkę do osobnych komór pojemnika na odpady. Po tygodniu przychodzi osoba przeprowadzająca badanie i wrzuca wszystko do jednego worka, mówiąc: "Wrzucam wszystko do jednego worka, zgodnie z procedurą"

*Appendix A.3. Manipulation Check*

A univariate analysis of variance showed a significant effect of the meaninglessness manipulation on perceived meaningfulness of the assistant's behavior, $F(1, 237) = 42.51$, $p < 0.001$, $\eta_p^2 = 15\%$, with persons in the meaningful condition ($M = 3.46$, $SD = 1.41$) finding meaning more than in the meaningless one ($M = 2.30$, $SD = 1.33$). We found no main effect of effort $F(1, 238) = 0.26$, $p = 0.611$, $\eta_p^2 = 0\%$ or the interaction $F(1, 238) = 0.32$, $p = 0.570$, $\eta_p^2 = 0\%$.

*Appendix A.4. Effect of Meaningless Effort on Disappointment*

An omnibus univariate analysis of variance with disappointment as a dependent variable yielded a significant main effect of the meaninglessness of the situation, $F(1, 233) = 107.24$, $p < 0.001$, $\eta_p^2 = 31\%$. Participants experienced more disappointment in the meaningless ($M = 3.66$, $SD = 1.08$) compared to the meaningful condition ($M = 2.24$, $SD = 1.09$). The main effect of effort was nonsignificant $F(1, 233) = 3.23$, $p = 0.073$, $\eta_p^2 = 1\%$. As predicted, the level of effort moderated the effect of the meaningfulness of the situation on disappointment, $F(1, 233) = 4.67$, $p < 0.05$, $\eta_p^2 = 2\%$. Means associated with the interaction are shown in Figure A1. Additionally, as men and women differed in the level of reported disappointment, we ran the same analysis controlling for gender as a covariate. ANCOVA showed that: (1) gender was not a significant covariate, $F(1, 233) = 3.72$, $p = 0.055$, $\eta_p^2 = 2\%$; and (2) the results were not changed significantly by including gender. Participants experienced more disappointment in the meaningless compared to the meaningful condition, $F(1, 233) = 104.85$, $p < 0.001$, $\eta_p^2 = 31\%$. The main effect of effort was nonsignificant $F(1, 233) = 2.33$, $p = 0.128$, $\eta_p^2 = 1\%$. The level of effort moderated the effect of the meaningfulness of the situation on disappointment, $F(1, 233) = 4.18$, $p < 0.05$, $\eta_p^2 = 2\%$.

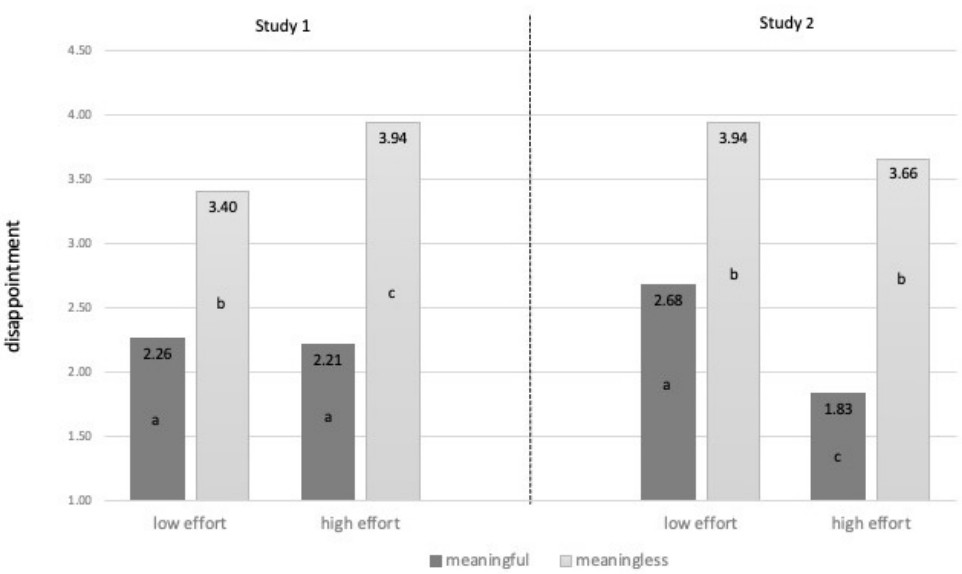

**Figure A1.** Effects of the manipulated meaninglessness of the situation and effort required to complete the task on disappointment (Studies 1 and 2). Note that means having different letter indexes differ significantly from each other.

*Appendix A.5. Sensitivity Analysis*

We ran a sensitivity analysis to test whether the moderated mediation model holds after including general negative emotions. We performed the moderated mediation Model 7 of the PROCESS macro with meaninglessness manipulation as an antecedent variable (X), sense in continuing as an outcome variable (Y) and disappointment as a mediator (M1). Additionally, we entered index of negative emotions as the second mediator (M2). This analysis yielded a significant index of moderated mediation for disappointment (0.08, $se = 0.04$, 95%$CI$ 0.006, 0.18), but the nonsignificant index of moderated mediation for negative emotions (0.03, $se = 0.03$, 95%$CI$ −0.017, 0.10). The results show that disappointment mediates the effect beyond and above other relevant emotions.

**Appendix B. Study 2**

The following are presented in this Appendix: (Appendix B.1) manipulation check; (Appendix B.2) effect of meaningless effort on disappointment; and (Appendix B.3) sensitivity analysis.

*Appendix B.1. Manipulation Check*

Next, we performed the ANOVA in which we included the index of difficulty. The analysis yielded a nonsignificant main effect of effort manipulation, $F(1, 282) = 4.22$, $p < 0.05$, $\eta_p^2 = 2\%$, in the expected direction. That is, in the high effort condition participants evaluated the task as more demanding ($M = 2.39$, $SD = 0.86$) than in the low-effort condition ($M = 2.61$, $SD = 0.94$). The main effect of meaninglessness manipulation, $F(1, 282) = 0.14$, $p = 0.705$, $\eta_p^2 = 0\%$, and the interaction were not significant, $F(1, 282) = 0.16$, $p = 0.900$, $\eta_p^2 = 0\%$.

*Appendix B.2. Effect of Meaningless Effort on Disappointment*

An omnibus univariate analysis of variance with disappointment as a dependent variable yielded a significant main effect of the meaninglessness of the situation, $F(1, 282) = 123.32$, $p < 0.001$, $\eta_p^2 = 31\%$. Participants experienced more disappointment in the meaningless ($M = 3.78$, $SD = 1.17$) compared to the meaningful condition ($M = 2.18$, $SD = 1.20$). The main effect of effort was also significant, $F(1, 282) = 16.60$, $p < 0.001$, $\eta_p^2 = 6\%$. Participants experienced less disappointment in the low-effort condition ($M = 2.72$, $SD = 1.49$)

compared to the effortful condition ($M$ = 3.34, $SD$ = 1.27). The level of effort moderated the effect of the meaningfulness of the situation on disappointment, $F(1, 282)$ = 4.27, $p < 0.05$, $\eta_p^2$ = 2%. The interaction was driven be the low experienced disappointment in the meaningful and high-effort condition and no difference between the means in the meaningless low effort and high effort condition (please refer to Figure A1).

*Appendix B.3. Sensitivity Analysis*

As in Study 1, we performed a sensitivity analysis of the model. Specifically, we performed two additional analyses with negative emotions and regret as additional mediators. First, we performed Model 7 of the PROCESS macro with meaninglessness manipulation as an antecedent variable (X), intention to continue the behavior as an outcome variable (Y) and disappointment as a mediator (M1). Additionally, we entered index of negative emotions as the second mediator (M2). This analysis yielded no significant index of moderated mediation for disappointment (0.01, $se$ = 0.02, 95%$CI$ −0.024, 0.075), but the significant index of moderated mediation for negative emotions (0.06, $se$ = 0.03, 95%$CI$ 0.001, 0.148). Second, we performed one more analysis, this time entering disappointment and regret as two mediators. This analysis yielded a significant index of moderated mediation for disappointment (0.05, $se$ = 0.03, 95%$CI$ 0.001, 0.12), but the nonsignificant index of moderated mediation for regret (0.01, $se$ = 0.02, 95%$CI$ −0.021, 0.051).

## Appendix C. Study 3

The following are presented in this Appendix: Appendix C.1 procedure of the study; Appendix C.2 instructions translated into English; Appendix C.3 pilot study and manipulation check; Appendix C.4 effect of meaningless effort on disappointment; Appendix C.5 effect of meaningless effort on conservation behavior; and Appendix C.6 sensitivity analysis.

*Appendix C.1. Procedure of the Study*

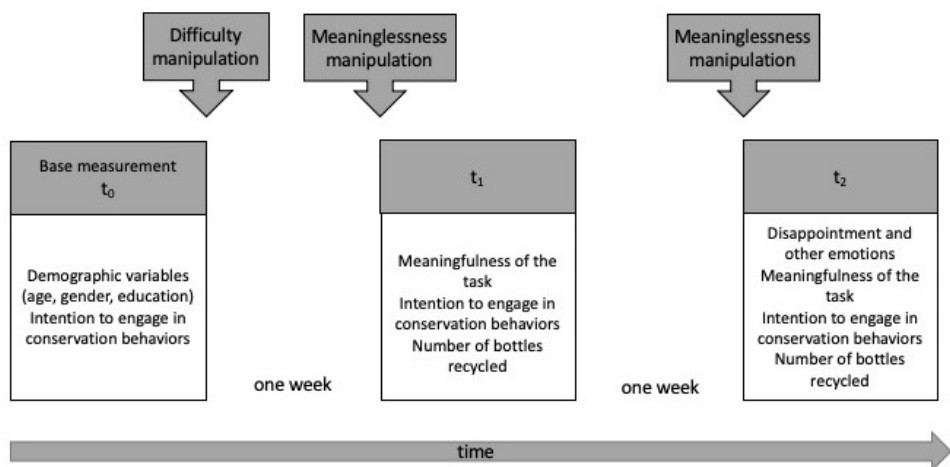

**Figure A2.** Flowchart of Study 3.

*Appendix C.2. Abstract from Instructions for Assistants Translated into English*

The Polish version of instructions is available on request.

### Group: meaningful/low effort

MEETING 1: Today I would like to give you water bottles and a container for segregation. Please, use the water from the bottles for your own needs, use the container for two weeks and separate the caps and bottles separately, and crush the bottles before throwing

them into the container chamber. AND YOU DON'T HAVE TO DO ANY MORE WITH THEM. I will come in a week and give you another batch of bottles. Please, remember that this is an individual study. It is extremely important that you only use the water for yourself and that you also sort the bottles and caps yourself. Please, do not throw any other waste into the bin.

MEETING 2: "Now I would like to hand over new bottles—please continue to sort the bottles after drinking the water from us for the next week, that is, please separate the caps and bottles separately and crush the bottles before throwing them into the container chamber."

"And now I'm going to pick up the old bottles."

In front of the examined person, we throw bottles separately, caps into two separate bags and say: "According to the instructions I received, I now throw individual types of waste into separate bags."

MEETING 3: "And now I'm going to pick up the old bottles."

In front of the examined person, we throw bottles separately, caps into two separate bags and say: "According to the instructions I received, I now throw individual types of waste into separate bags."

### Group: meaningless/low effort

MEETING 1: Today I would like to give you water bottles and a container for segregation. Please, use the water from the bottles for your own needs, use the container for two weeks and separate the caps and bottles separately, and crush the bottles before throwing them into the container chamber. AND YOU DON'T HAVE TO DO ANY MORE WITH THEM. I will come in a week and give you another batch of bottles. Please, remember that this is an individual study. It is extremely important that you only use the water for yourself and that you also sort the bottles and caps yourself. Please, do not throw any other waste into the bins.

MEETING 2: "Now I would like to hand over new bottles—please continue to sort the bottles after drinking the water from us for the next week, that is, please separate the caps and bottles separately and crush the bottles before throwing them into the container chamber."

"And now I'm going to pick up the old bottles."

In front of the examined person, we throw everything into one bag and say: "According to the instructions I received, I am now throwing everything into one bag."

MEETING 3: "And now I'm going to pick up the old bottles."

In front of the examined person, we throw everything into one bag and say: "According to the instructions I received, I am now throwing everything into one bag."

### Group: meaningful/high effort

MEETING 1: Today I would like to give you water bottles and a container for segregation. Please, use the water from the bottles for your own needs, use the container for two weeks and segregate the bottles—removing the labels, crushing the bottles and throwing the caps separately, the bottles separately and the labels to the individual chambers of the container. I will come in a week and give you another batch of bottles. Please, remember that this is an individual examination. It is extremely important that you only use the water for yourself and that you also sort the bottles and caps yourself. Please, do not throw any other waste into the bins.

MEETING 2: "Now I would like to hand over new bottles—please, continue sorting the bottles after drinking the water from us for the next week, that is, take the labels off, crush the bottles and throw out the caps, bottles and labels separately to each compartment of the container. I would also like to confirm the date of our meeting which we agreed recently. Then I will pick up the containers and give a gift."

"And now I'm going to pick up the old bottles."

In front of the examined person, we throw bottles, caps separately, labels separately into three separate bags and say: "According to the instructions I received, I am now throwing individual types of waste into separate bags."

MEETING 3: "And now I'm going to pick up the old bottles."

In front of the examined person, we throw bottles, caps separately, labels separately into three separate bags and say: "According to the instructions I received, I am now throwing individual types of waste into separate bags."

**Group: meaningless/high effort**

MEETING 1: Today, I would like to give you water bottles and a container for segregation. Please, use the water from the bottles for your own needs, use the container for two weeks and segregate the bottles—removing the labels, crushing the bottles and throwing the caps separately, the bottles separately and the labels to the individual chambers of the container. I will come in a week and give you another batch of bottles. Please, remember that this is an individual study. It is extremely important that you only use the water for yourself and that you also sort the bottles and caps yourself. Please, do not throw any other waste into the bins.

MEETING 2: "And now I'm going to pick up the old bottles."

In front of the examined person, we throw everything into one bag and say: "According to the instructions I received, I am now throwing everything into one bag."

MEETING 3: In front of the examined person, we throw everything into one bag and say: "According to the instructions I received, I am now throwing everything into one bag."

*Appendix C.3. Pilot Study and Manipulation Check*

A one-way analysis of variance showed a significant effect of the meaninglessness manipulation on searching for meaning, $F(1, 271) = 3.95$, $p < 0.05$, $\eta_p^2 = 2\%$, with persons in the meaningless condition ($M = 3.46$, $SD = 1.36$) searching for meaning more than those in the meaningful condition ($M = 3.11$, $SD = 1.44$). We found, however, no effect of the manipulation on the perceived meaningfulness of the task, $F(1, 268) = 0.13$, $p = 0.716$, $\eta_p^2 = 0\%$.

As for the effort, past research has shown that subjective and objective difficulties of behaviors are not perfectly related (see, e.g., [54]). Therefore, to test the manipulation of effort based on the difficulty of the task instead of relying exclusively on people's perceptions, we additionally measured the time required to complete the task in the easy and the difficult condition. Prior to Study 3, we conducted a pilot study in a laboratory in which 60 participants (mean age: $M = 22.80$, $SD = 4.67$; 66% female) were randomly allocated to a low effort (0) or high effort (1) condition and were asked to recycle five bottles by putting them into a rotary bin, which was later used in the main study. We found the expected differences between the groups in terms of the time required to complete the task in the easy ($M = 0.04$, $SD = 0.05$) and the difficult ($M = 0.19$, $SD = 0.05$) condition measured in seconds, $F(1, 58) = 228.48$; $p < 0.001$, $\eta_p^2 = 80\%$. The difficult task was also perceived as more laborious, $F(1, 59) = 8.01$, $p < 0.05$, $\eta_p^2 = 12\%$; burdensome, $F(1, 59) = 4.64$; $p < 0.05$, $\eta_p^2 = 7\%$; and troublesome, $F(1, 59) = 11.35$, $p < 0.05$, $\eta_p^2 = 16\%$.

*Appendix C.4. Effect of Meaningless Effort on Disappointment*

A univariate analysis of variance with disappointment as a dependent variable yielded no significant main effect of the manipulation of the meaninglessness of the situation, $F(1, 263) = 0.04$, $p = 0.840$, $\eta_p^2 = 0\%$, nor a main effect of effort, $F(1, 263) = 0.95$, $p = 0.332$, $\eta_p^2 = 0\%$, or interaction, $F(1, 263) = 0.11$, $p = 0.736$, $\eta_p^2 = 0\%$.

As previous studies and the results of our manipulation check suggest that people may attribute subjective meanings to situations, we performed an analysis in which we replaced the meaninglessness condition with a continuous variable in which participants evaluated whether the task made sense to them. We tested expected moderation using Model 1 of the PROCESS macro [37] by entering the perceived meaningfulness of the task ($t_2$) as a predictor and the effort condition (low effort = 0 vs. high effort = 1) and disappointment

($t_2$) as an outcome variable. All variables were standardized and mean-centered before the analysis to minimize multicollinearity.

We found the main effect of the perceived meaningfulness ($\beta = -0.17, t = -2.71,$ $p < 0.05, 95\%CI - 0.29, -0.05$) but no significant main effect of the effort ($\beta = 0.06, t = 0.94,$ $p = 0.349, 95\%CI - 0.06, 0.18$). We also found the expected significant interaction ($\beta = -0.17,$ $t = -2.62, p < 0.05, 95\%CI - 0.29, -0.04$). As Figure 4 shows, closer inspection of the results revealed that participants experienced more disappointment if they were in the effortful condition and perceived the task as less meaningful. We entered gender and age into the model, but it did not affect the main findings. The conditional effects analysis showed that, in the effortful condition, perceived meaningfulness of the task was related to disappointment ($\beta = -0.32, t = -4, 21, p < 0.001, 95\%CI - 0.46, -0.17$), but it was not in the low-effort condition ($\beta = 0.02, t = 1.63, p = 0.871, 95\%CI - 0.19, 0.21$).

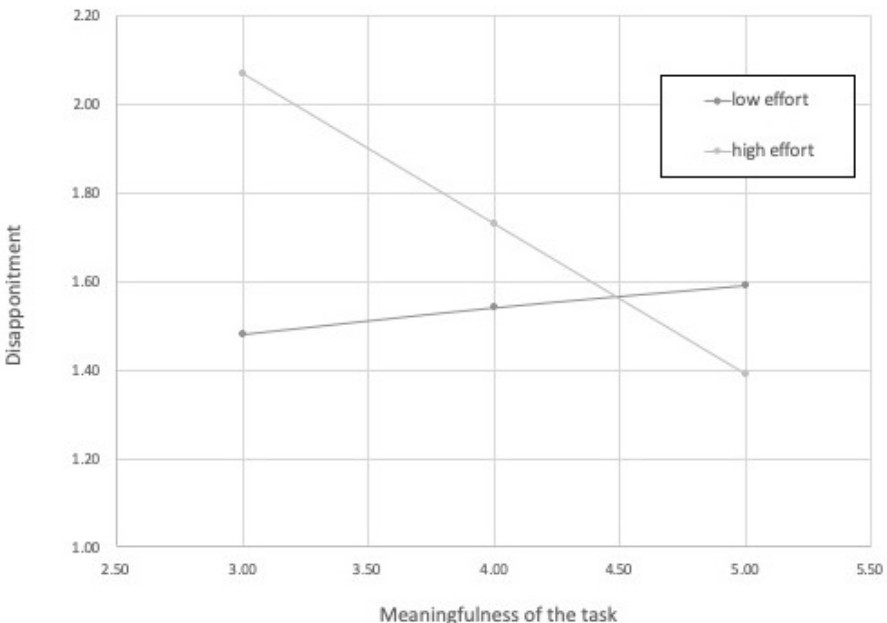

**Figure A3.** Effects of the perceived meaningfulness of the situation on disappointment moderated by the amount of effort required to complete task.

*Appendix C.5. Effect of Meaningless Effort on Conservation Behavior*

A mixed-effects analysis of variance of the mean number of bottles recycled per day as a within-subject variable yielded no main effect of the time of measurement, $F(1, 265) = 0.84, p = 0.361, \eta_p^2 = 0\%$, the meaninglessness of the situation, $F(1, 265) = 0.03,$ $p = 0.862, \eta_p^2 = 0\%$, the difficulty $F(1, 265) = 0.89, p = 0.347, \eta_p^2 = 0\%$, and no interaction, $F(1, 265) = 0.01, p = 0.912, \eta_p^2 = 0\%$. We found no other two-way interactions of the time of measurement × the meaninglessness of the situation, $F(1, 265) = 0.16, p = 0.687$, or the time of measurement × the difficulty of the situation, $F(1, 265) = 0.19, p = 0.663, \eta_p^2 = 0\%$. The three-way interaction of the time of measurement, meaningfulness and difficulty was also nonsignificant, $F(1, 265) = 0.00, p = 0.993, \eta_p^2 = 0\%$.

Using the same logic as in the analysis with disappointment, we performed moderation using Model 1 of the PROCESS macro [37]. We entered the perceived meaningfulness of the task ($t_2$) as a predictor, the effort condition (low effort = 0 vs. high effort = 1) as a moderator and the number of bottles recycled after the manipulation ($t_2$) as an outcome variable, controlling for the number of bottles recycled before the manipulation ($t_1$). All variables were mean-centered and standardized before the analysis. We found no main effect of the perceived meaningfulness ($\beta = -0.05, t = -0.82, p = 0.413, 95\%CI - 0.17, 0.07$) and no significant main effect of effort ($\beta = -0.05, t = -0.76, p = 0.448, 95\%CI - 0.17, 0.07$). The

expected interaction was nonsignificant at a *p*-value of 0.05, but close to the customary cut-off point ($\beta = -0.11, t = -1.74, p = 0.083$).

*Appendix C.6. Sensitivity Analysis*

We ran a sensitivity analysis with meaninglessness manipulation as an antecedent variable (X), intention to engage in conservation behaviors in the future as an outcome variable (Y) and disappointment as a mediator (M1). Additionally, we entered index of negative emotions as the second mediator (M2). This analysis yielded no significant index of moderated mediation for disappointment ($0.05, se = 0.04, 95\%CI - 0.01, 0.16$) and the nonsignificant index of moderated mediation for negative emotions ($0.005, se = 0.03, 95\%CI - 0.057, 0.072$). The effect of disappointment on intention when controlling for negative emotions was marginally nonsignificant ($\beta = -0.13, t = -1.60, p = 0.11, 95\%CI - 0.29, 0.029$). We also ran the same model entering positive emotions as the second mediator (M2) instead of negative ones. This analysis yielded a significant index of moderated mediation for disappointment ($0.06, se = 0.03, 95\%CI\ 0.003, 0.14$) and the nonsignificant index of moderated mediation for positive emotions ($-0.01, se = 0.01, 95\%CI - 0.04, 0.02$). The effect of disappointment on intention when controlling for positive emotions remained significant ($\beta = -0.14, t = -2.07, p = 0.05, 95\%CI - 0.28, -0.01$).

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
