# Peer review of "Consequences of Sisyphean Efforts: Meaningless Effort Decreases Motivation to Engage in Subsequent Conservation Behaviors through Disappointment"

_sustainability, doi:10.3390/su13105716_

Round 1

Reviewer 1 Report

Introduction: 

Line 24 "efforts that do not make sense" - please try to be more neutral, this statement inherit some judgement. To whom, I could ask, do theses efforts make no sense? Certainly for those perfoming it, it really makes sense.

How do you define "a meaningfulness or meaningless task"? What are the psychology behind such interpretations? What about individualized perceptions?

Your focus is to understand and research about the outpout feelings of such tasks. What about the insight each one has regarding those tasks and how they link to certain life events? 

Study 1:

Which statsitical software was used for the analysis?

Is it possible to report together with p-values, effect sizes? For example at lines 256 and 257.

Study 2:

Are the participants in this study independent from study 1? Is there any repeated measure regarding participation?

Line 376 - Appendix reference must be corrected.

Study 3:

After deleting missing values what was the total sample size used for the analysis? If the missing data was MCAR, why did the authors not performe multiple imputation?

Line 552 - correct the Appendix reference

Line 552: how is disappointment positively related to willingness to engange with a negative beta estimator?

Figure 4: How is intention t0 influencing the other variables? No reference values or whatsoever was displayed.

Author Response

Best regards,

Dr. Byrka

Reviewer 2 Report

The authors describe a series of experiments that examine low or high effort engagement in activities that are perceived as meaningless, and the impact that subsequent disappointment has on behavioural intentions. While some of these effects have been demonstrated in other studies, the authors note that this is the first study to examine the interaction between effort and emotion in conservation behaviours that are perceived as meaningless. It is an interesting proposal, but the work has technical limitations and the effects are small (especially for Study 3) that mean its implications are difficult to extrapolate to any real-world sustainability outcomes. I would also like to see a deeper exploration of why this matters for conservation behaviour, specifically. This could be a study about any behaviour, but in submitting it to this journal, I would expect to see a more in-depth discussion of the implications for sustainability.

Below I outline some specific comments on the various sections of the manuscript and a brief summary of my overall assessment.

INTRO/AIMS & OBJECTIVES

Lines 15-17: I am not sure I quite understand the observation on which the study is based. What, in this context is a “track”. Where did these observations take place?

It’s an unusual way to start a paper, which is not a problem per se, but if the authors want to use ‘a hook’ in this way, I think they need to elaborate on this crucial observation.

Line 31 – Could the authors provide a reference? Where they say “circumstantial evidence”, what do they mean?

Lines 44-51: If I understand correctly, the hypothesis is that people are less likely to engage in conservation behaviours if they experience disappointment because the activity they were engaged in before is perceived to be meaningless? Given the fact that there is a fair amount of evidence linking meaninglessness to reduced motivation, is the novel facet of this study just the inclusion of a mediating factor, i.e. feelings of disappointment? I think if the authors want to emphasise the novelty of the study, it should be on the basis that they are applying a tested paradigm to the study of conservation behaviours specifically. That said, it seems to me that this principle would apply to any behaviour. I am left feeling uncertain of what the contribution of this paper is.

Sections 1.1. and 1.2. – I’m not sure I understand the difference between meaninglessness and meaningless effort. For instance, in section 1.1. the authors reference multiple experiments were participants are apparently engaged in meaningless efforts (e.g. building robots that were destroyed, or marking infected cells in images).

Line 102: “[…] when they see no relations between their actions.” Should this be “[…] actions and their outcomes”?

Lines 112-113: I don’t understand this sentence, or the example. A football game is never a meaningless task, because it probably involves a range of inherent meanings and desired outcomes (social engagement, physical exercise) beyond just the end result of the actual game (win or lose). Maybe the authors need to think of a different example, because this one confounds a number of issues.

Lines 182-183: Is “sense in continuing” the behavioural intention of participants? What is meant with “sense”? Is the question whether it is “rational” or whether it is “effective”?

STUDY 1

Can the authors include some more detail about the origin of the sample? Are they university students? What field? Did they receive an incentive to take part in the study?

It is an entirely hypothetical scenario. Notably, the women in the sample were found to experience more disappointment. This could just be because they were more empathetic? This begs the question whether this experiment is really a measure of disappointment, or is this a measure of empathic skill? The authors should address this.

Line 243 – 244: The question is not a metric of the meaningfulness of the task itself? Should the question not be about the participants’ behaviour instead of the research assistant?

Lines 280-282: Could the authors rephrase? It’s unclear. What is meant with “theoretically justified”?

There is a typo in Figure 1 (“MNIPULATION”)

STUDY 2

The same applies with respect to the sample description. Some more detail would be useful. I also don’t think the gender balance was equivalent – there were twice as many women as men.

Line 376: correct the reference to the appendix.

The disappearance of the mediation effect when other negative emotions were included makes me wonder how specific the effect is to disappointment. Can we trust this effect in Study 1? The very strong correlation between the negative emotions suggests that negative emotions in general mediate the effect, not just disappointment alone. It would have been interesting if the authors had included positive emotions as well – as a control, if you will.

STUDY 3

This study obviously addresses the limitations of the hypothetical scenarios used in Study1-2. It’s an impressive feat to conduct this field experiment with N=286 completing all three timepoints.

However, I did wonder to what extent the participants were aware of the actual objective of the study? Could this have influenced their behavioural intentions? Especially since the actual behaviour did not change, but their reported intentions did – it could be influenced by demand effects? Could the authors discuss this?

SUMMARY/GENERAL COMMENTS

To be interesting to the readership of sustainability, the authors need to address the issue of conservation behaviour more broadly. At the moment, this is not a study of conservation behaviour, this is more fundamentally an experiment about individuals’ perception of meaning, effort, and emotions elicited by a task. It is probably not specific to conservation behaviour, but perhaps any pro-social behaviour or even self-serving behaviour, although perhaps patterns of emotional engagement/disappointment may vary. What I mean is that this study requires more context – how does this finding help us understand, change or support conservation behaviours more specifically?

It might be difficult to answer that questions, because fundamentally, this is not actually a study about behaviour per se, but about people’s perception about hypothetical behaviour. Even in study 3, the behavioural measurement itself might have been too simple and too specific? What is more interesting is the effect of disappointment and perceptions of meaninglessness on general conservation behavioural intentions that are less transparently being manipulated in the experiment itself. However, the effect was so small, that it does beg the question whether the it is useful in practical sense. In terms of achieving sustainability outcomes – does any of this really matter? How could we meaningfully quantify this?

Author Response

Best regards,

Dr. Byrka

Round 2

Reviewer 2 Report

I commend the authors on their constructive engagement with the review process. I do feel that the paper has improved, but I still feel that the results are at best "mildly suggestive" that behavioural intentions are affected by the perceived meaningless of actions. The key test lies in Experiment 3, where the theory is tested under more realistic conditions and extended beyond the direct conservation behaviour (recycling). The results of this particular study are underwhelming (see Figure 3). I don't disagree with the authors that small individual changes, on a large scale (given the widespread use of recycling) can be impactful. However, it remains difficult to argue that these very minor differences in behavioural intentions will actually impact behaviour, let alone sustainability outcomes, as the study provided no direct evidence of any behaviour change. 

The findings may inspire more research on the role of affect/emotion in pro-environmental behaviour. That is an area that is undoubtedly under-researched. However, given the very modest findings, the authors may wish to focus on clearer recommendations on how we might advance this area of research. As it stands, I fear the conclusions are not entirely supported by the results and a more cautious approach is warranted.

The writing is of a good standard, but I would recommend that the authors ask a native English speaker to proofread the final version.

Please see the attached file for some additional in-text comments.

Author Response

Thank you for such a thorough review of our manuscript. We agree with the Reviewer that our study does not provide evidence for changes in behaviors. We revised the General Discussion accordingly. In line with the Reviewer's comment, we also suggest in which direction research on affective triggers of behavior change could be developed.

We are grateful for all specific comments in the PDF file. We corrected everything as pointed. The new text has been reviewed once again by a native speaker.

All changes in the text made in R2 are in orange in the text.

Once again thank you for devoting time to this review.

Sincerely yours,

Katarzyna Byrka